# Core Fucosylation Mediated by the FucT-8 Enzyme Affects TRAIL-Induced Apoptosis and Sensitivity to Chemotherapy in Human SW480 and SW620 Colorectal Cancer Cells

**DOI:** 10.3390/ijms241511879

**Published:** 2023-07-25

**Authors:** Rubén López-Cortés, Isabel Correa Pardo, Laura Muinelo-Romay, Almudena Fernández-Briera, Emilio Gil-Martín

**Affiliations:** 1Doctoral Program in Methods and Applications in Life Sciences, Faculty of Biology, Universidade de Vigo, Campus Lagoas-Marcosende, ES36310 Vigo, Spain; rlcortes.eu@gmail.com; 2Master Program in Advanced Biotechnology, Faculty of Biology, Universidade de Vigo, Campus Lagoas-Marcosende, ES36310 Vigo, Spain; icorreap@uvigo.es; 3Liquid Biopsy Analysis Unit, Translational Medical Oncology (Oncomet), Health Research Institute of Santiago de Compostela (IDIS), CIBERONC, Travesía da Choupana, ES15706 Santiago de Compostela, Spain; lmuirom@gmail.com; 4Department of Biochemistry, Genetics and Immunology, Faculty of Biology, Universidade de Vigo, Campus Lagoas-Marcosende, ES36310 Vigo, Spain; abriera.fernandez@gmail.com

**Keywords:** *FUT8* knockdown, core fucosylation, TRAIL-induced apoptosis, DR4, colorectal cancer

## Abstract

Epithelial cells can undergo apoptosis by manipulating the balance between pro-survival and apoptotic signals. In this work, we show that TRAIL-induced apoptosis can be differentially regulated by the expression of α(1,6)fucosyltransferase (FucT-8), the only enzyme in mammals that transfers the α(1,6)fucose residue to the pentasaccharide core of complex N-glycans. Specifically, in the cellular model of colorectal cancer (CRC) progression formed using the human syngeneic lines SW480 and SW620, knockdown of the FucT-8-encoding *FUT8* gene significantly enhanced TRAIL-induced apoptosis in SW480 cells. However, *FUT8* repression did not affect SW620 cells, which suggests that core fucosylation differentiates TRAIL-sensitive premetastatic SW480 cells from TRAIL-resistant metastatic SW620 cells. In this regard, we provide evidence that phosphorylation of ERK1/2 kinases can dynamically regulate TRAIL-dependent apoptosis and that core fucosylation can control the ERK/MAPK pro-survival pathway in which SW480 and SW620 cells participate. Moreover, the depletion of core fucosylation sensitises primary tumour SW480 cells to the combination of TRAIL and low doses of 5-FU, oxaliplatin, irinotecan, or mitomycin C. In contrast, a combination of TRAIL and oxaliplatin, irinotecan, or bevacizumab reinforces resistance of *FUT8*-knockdown metastatic SW620 cells to apoptosis. Consequently, FucT-8 could be a plausible target for increasing apoptosis and drug response in early CRC.

## 1. Introduction

The tumour necrosis factor (TNF)-related apoptosis-inducing ligand (TRAIL) is a member of the TNF family that induces regulated cell death through caspase-dependent apoptosis in tumours of diverse origins, while sparing most normal cells [1,2,3]. This ability of TRAIL to discriminate between healthy and malignant cells has enabled clinical trials to test soluble recombinant versions of TRAIL and agonistic antibodies [4] in patients affected by different cancers, with promising results for the cytokine in the selective destruction of malignant cells [5,6,7].

TRAIL regulates apoptosis by binding to four different transmembrane-anchored cell surface receptors. TRAIL-R1 (death receptor 4, DR4) and TRAIL-R2 (death receptor 5, DR5) induce programmed cell death, while antagonistic receptors TRAIL-R3 (decoy receptor 1, DcR1) and TRAIL-R4 (decoy receptor 2, DcR2) fail to induce apoptosis as they lack the functional cytoplasmic death domain (DD). In addition, TRAIL binds to osteoprotegerin (OPG), a soluble decoy receptor that also recognises the nuclear factor kappa B ligand (RANKL), thus displaying antiapoptotic and proliferative activity. After TRAIL binds with OPG, DR4 and DR5 oligomerise as their cytosolic DDs cluster. Next, DR4 and DR5 engage with the Fas-associated death domain (FADD) adaptor and pro-caspase 8 via homotypic interactions to assemble the death-inducing signalling complex (DISC) [8,9]. Afterwards, proteolytic trimming of caspase 8 triggers the subsequent cleavage of effector caspases 3, 6, and 7, which completes the canonical extrinsic or type I apoptosis pathway. Alternatively, the intrinsic or type II pathway, also known as mitochondrial-regulated apoptosis, depends on mitochondrial integrity and typically begins with endogenous insults (e.g., DNA damage produced by chemotherapy or radiation) [10], although extracellular ligands such as TRAIL can also trigger the pathway [11]. In the intrinsic apoptosis dimerization of proapoptotic B cell lymphoma-2 (Bcl-2)-family proteins, Bcl-2-associated X protein (BAX) and Bcl-2 homologous antagonist killer (BAK) make the outer membrane permeable and disrupt mitochondria, releasing cytochrome c or the second mitochondria-derived activator of caspases (SMAC/DIABLO) into the cytosol to initiate the caspase cascade via caspase 9 [12].

In addition to activating and inhibitory receptors, TRAIL-mediated apoptosis is controlled by a myriad of proapoptotic and antiapoptotic cytosolic mediators [13], as well as signalling pathways that can amplify or even suppress apoptosis [14], suggesting that TRAIL plays a role physiologically more complex than that of a mere “death bullet” ligand. The global balance of apoptotic and non-apoptotic pro-survival signals is pivotal in determining the point of no return for cell death, allowing sensitivity towards apoptosis and cell fate to be regulated at different levels. In this regard, protein-kinase-mediated phosphorylation-based cascades are key modulation targets that have garnered interest for providing rapid responses independent of the synthesis of new protein [15]. The case of the mitogen-activated protein kinase/extracellular signal-regulated kinase (MAPK/ERK) pathway is paradigmatic because it mediates the transduction of mitogens that drive cell growth, survival, and differentiation [16], as shown by the inhibition of caspase-induced apoptosis promoted by pERK1/2 kinase through the direct phosphorylation of pro-caspase 8 [17]. Since a lack of apoptosis may result in tumour resistance and relapse, more research on the interaction of TRAIL with pro-survival and apoptotic pathways is crucial to improve the effectiveness of anticancer therapy.

It has been postulated that around 70% of human proteins undergo post-transcriptional multi-step glycosylation [18]. Therefore, in molecular oncology, they attract a lot of attention as potential key mediators of carcinogenesis. Indeed, increased fucosylation and sialylation or truncated O-glycosylation are characteristic glycan signatures of tumour cell dyshomeostasis [19]. Particularly, the importance of fucosylated glycoproteins involved in signal transduction, cell growth, and plasticity, as well as cell-to-cell and cell-to-matrix adhesion [20,21], has positioned aberrant fucosylation as one of the hallmarks of malignant transformation [22]. Furthermore, glycosylation is also involved in tumour apoptosis, as suggested by the N-glycosylation sequence (Asn-X-Ser/Thr) at Asn156 of DR4 [23] and O-glycosylation sites harboured by DR5 [24], which could underpin its mechanistic involvement in TRAIL:death-receptor binding and/or receptor activation. Thus, it has been reported that the expression of fucosyltransferases FucT-1, FucT-3, or FucT-6 or the fucose synthesis enzyme GDP-mannose-4,6-dehydratase (GMDS), as well as the methylation of fucosylation genes, may affect TRAIL-induced apoptosis [24,25,26].

The transfer of α(1,6)fucose from GDP-β-L-fucose to the innermost GlcNAc residue of the pentasaccharide core of Asn-linked glycopeptides is called α(1,6)fucosylation, usually referred to as core fucosylation. In mammals, the single Golgi-resident enzyme α(1,6)fucosyltransferase (FucT-8, EC 2.4.1.68) accomplishes this essential glycosylation modality of eukaryotic complex N-glycans [27,28,29]. Recent structural information has indicated that core fucosylation is a determinant of N-glycoprotein functionality on the basis that conformation [30] and stereoselective interactions with specific partners [31] are partially conditioned with core α(1,6)fucose labelling [32]. Moreover, compelling evidence has linked core fucosylation at specific glycosites of signalling proteins, such as growth factor EGFR, TGFR-β1, or VEGFR-2, with tumour development in the same manner as that of core-fucosylated species, such as adhesion receptor E-cadherin or α3β1 and α5β1 integrins, which have been associated with the spread of metastatic cells [20]. As a result, α(1,6)fucosylated proteins have attracted a lot of attention as reliable cancer biomarkers and therapeutic targets [33]. It is important to note that DR4 is a promising target for the regulation of apoptosis, and presumably the death receptor undergoing core fucosylation [23,34], so FucT-8 may be involved in the resistance of cancer cells to death. Additionally, P-glycoprotein (PGP), an efflux pump responsible for extruding anti-tumour drugs from the cytoplasm of multidrug-resistant cells [35,36], particularly colorectal cancer (CRC) cells [37], is also susceptible to being α(1,6)fucosylated [38]. PGP has aroused great translational interest in clinical trials designed to re-sensitise cancer cells to chemotherapy [39], elucidating that PGP’s dependence on FucT-8 may help circumvent multidrug resistance.

As per available data, core fucosylation plays an active role in the escape of colorectal cells from the homeostatic control that determines exacerbated proliferation and malignant progression of colorectal cells [20,40,41]. Taking into account that understanding how tumour cells avoid programmed cell death or elude radio/chemotherapy cytotoxicity may open new avenues for therapy, we decided to analyse the impact of inhibiting core fucosylation on TRAIL-induced apoptosis. For this purpose, we used the cellular model of CRC progression composed of the isogenic lines SW480 (low metastatic) and SW620 (highly metastatic), from which we selected two FucT-8-deficient clones per line, obtained through stable knockdown with specific lentiviral shRNAs for the *FUT8* gene encoding FucT-8 [41]. Subsequently, *FUT8*-knockdown clones, their non-targeted controls (NTCs), and wild-type (wt) SW480 and SW620 cells were examined for TRAIL sensitivity, expression of FucT-8, and mediators of activation and regulation of extrinsic/intrinsic apoptosis, as well as growth inhibition achieved via chemotherapy drugs with or without TRAIL. The results showed increased TRAIL-induced apoptosis in *FUT8*-knockdown clones from SW480 primary tumour cells and, in this regard, provided evidence that core fucosylation and/or phosphorylation of ERK1/2 MAP kinases can modulate TRAIL signalling. In addition, the depletion of core fucosylation sensitised the SW480 line to the co-administration of TRAIL and chemotherapy drugs. In contrast, its metastatic counterpart SW620 was resistant to TRAIL-induced apoptosis and did not respond to *FUT8* knockdown. Taken together, as per our results, core fucosylation appears to be relevant in early CRC, where it may influence cell fate, progression, and response to treatment.

## 2. Results

### 2.1. TRAIL-Mediated Apoptosis Increased in SW480 Cells Inhibited for Core Fucosylation via FUT8 Gene Knockdown

We compared the ratio of TRAIL-mediated apoptosis in knockdown clones for *FUT8* mRNA (F52L and F59L) with respect to their corresponding transduction controls (NTC: non-targeted controls) and wild-type (wt) SW480 and SW620 colorectal cancer (CRC) cells. For this purpose, cells were seeded and allowed to form a monolayer for 24 h. Afterwards, they were exposed to 50, 100, or 200 ng/mL TRAIL and left to undergo apoptosis for 24 h. All these TRAIL concentrations produced similar percentages of cell death, with no significant differences between doses. According to the above, in Figure 1, we plotted the percentage of surviving wt SW480/SW620, NTC, and F52L/F59L cells at 100 ng/mL TRAIL. In the SW480 line, F52L and F59L knockdown clones were more sensitive to TRAIL than NTC, as only 26.01% ± 8.7 and 26.61% ± 5.6 of the F52L and F59L cells survived, respectively (*p* < 0.05 according to the Mann–Whitney U test; Figure 1A). In contrast, TRAIL sensitivity of the SW620 line was not affected by *FUT8* knockdown (Figure 1B). Moreover, metastatic SW620 cells showed greater overall resistance to TRAIL-dependent apoptosis than primary SW480 counterparts, as evidenced by their corresponding survival rates, i.e., 63% ± 1.8 in wt SW620 cells vs. 43% ± 5.7 in wt SW480 cells (Figure 1B).

### 2.2. Expression Levels of FucT-8 and DR4 or PGP Changed Inversely in the FUT8-Knockdown Clones of SW480 and SW620 Lines

Initially, we examined the expression profile of FucT-8 and DR4 in *FUT8*-knockdown cells (clones F52L and F59L), their corresponding transduction controls (NTC), and original wt cells. In this regard, FucT-8 expression was significantly attenuated in the *FUT8*-knockdown clones of SW480 (Figure 2A,B) and SW620 (Figure 2C,D) cells, as reported for *FUT8* mRNA (Appendix A), while DR4 displayed statistically significant upregulation (*p* < 0.05 according to the Mann–Whitney U test for comparisons of FucT-8 and DR4 expression of F52L/F59L and NTC clones (Figure 2B,D for lines SW480 and SW620, respectively). These immunoblot results were further validated using immunofluorescence assay (Figure 3A, SW480 cells; Figure 3C, SW620 cells), in which the same inverse evolution of FucT-8 and DR4 expression was observed (*p* < 0.05 according to the Mann–Whitney U test for comparisons of FucT-8 and DR4 levels in F52L/F59L-knockdown and NTC clones; See Figure 3B,D for lines SW480 and SW620, respectively).

PGP was included in the panel of proteins assayed using immunofluorescence because it is involved in the resistance of CRC cells to chemotherapy, and glycosylation influences its activity [36,37,42]. The results indicated an apparent overexpression of PGP in the *FUT8*-knockdown F52L and F59L clones from the SW480 line (Figure 3A,B) and the F52L clone from the SW620 line (Figure 3C,D). However, the PGP increment was not statistically significant, as quantification of the *ABCB1* gene using RT-qPCR subsequently confirmed (Appendix A).

### 2.3. Effects of TRAIL at 100 ng/mL on the Expression of FucT-8 and DR4 in SW480 and SW620 Colorectal Cancer Cells

After reporting the effect of *FUT8* knockdown on FucT-8 and DR4 expression, we decided to investigate the influence of treatment with 100 ng/mL TRAIL. Regarding DR4, TRAIL in the culture medium of SW480 cells maintained the upregulated expression of FucT-8 and DR4 produced in F52L/F59L clones through *FUT8* knockdown (Figure 4A; *p* < 0.05 according to the Mann–Whitney U test). However, in F52L/F59L clones from the SW620 line, TRAIL did not influence DR4 expression (Figure 4B). In addition, wt SW480 cells responded to the cytokine by increasing FucT-8 (Figure 4C; *p* < 0.05 according to the Mann–Whitney U test) to the point where their F52L and F59L clones showed partial reversal of the inhibition achieved through *FUT8* gene knockdown (Figure 4C; *p* < 0.05 according to the Mann–Whitney U test). In the SW620 line, FucT-8 was also upregulated in wt cells exposed to TRAIL (Figure 4D; *p* < 0.05 according to the Mann–Whitney U test), but their corresponding *FUT8*-knockdown F52L/F59L clones maintained unchanged FucT-8 expression compared to their non-TRAIL counterparts (where FucT-8 expression was reduced; *p* < 0.05 according to the Mann–Whitney U test for the reduced expression of FucT-8 in the presence or absence of TRAIL).

Unlike FucT-8, the knockdown of *FUT8* in SW480 (Appendix A) and SW620 (Appendix A) cells did not influence the expression levels of core fucosylated proteins. Contrarily, TRAIL elicited a significant increase of α(1,6)fucosylated species (*p* < 0.05 according to the Mann–Whitney U test) in wt SW480 cells (Appendix A) and SW480/SW620 NTC cells (Appendix A, respectively). This effect was not observed either in wt SW620 cells (Appendix A) or in *FUT8*-knockdown clones from both cell lines.

### 2.4. Increased TRAIL-Induced Apoptosis in FucT-8-Attenuated SW480 Cells Might Depend on Caspase 9

To confirm whether caspases enhance the TRAIL-induced apoptosis observed in SW480 F52L and F59L clones, we analysed the activation of the intrinsic and extrinsic pathways in the presence of 100 ng/mL TRAIL by comparing the expression of caspases (3, 6, 7–9) and PARP (poly-(ADP-ribose) polymerase) with that observed in basal (non-induced) conditions (Figure 5). Representative blots of these markers are included in Appendix A.

In the SW480 panel, TRAIL changed the pro-caspase 8/caspase 8 ratio of wt cells (basal conditions) to a new inverse ratio of isoforms in clones NTC, F52L, and F59L, indicative of protease activation, as seen in Figure 5A, by increasing the active cleaved isoform (cCasp8). However, the depletion of FucT-8 significantly moderated active cCasp8 in knockdown clones F52L and F59L in comparison with NTC (*p* < 0.05 according to the Mann–Whitney U test; Figure 5A). Similarly, TRAIL activated Casp9 in SW480 F52L and F59L clones, as indicated by the significant increase in the cCasp9 cleaved isoform (*p* < 0.05 according to the Mann–Whitney U test; Figure 5C). Similarly, the increase in cPARP indicated PARP activation in wt SW480 cells and their *FUT8*-attenuated clones (Figure 5K; *p* < 0.05 according to the Mann–Whitney test U).

However, in F52L and F59L clones of the SW620 line, *FUT8* knockdown significantly diminished cCasp8, while TRAIL significantly increased cCasp8 (*p* < 0.05 in both cases according to the Mann–Whitney U test; Figure 5B). However, pro-caspase 8 activation on TRAIL exposure was not driven downstream through the caspase 9 loop, as we did not detect the presence of the cleaved active cCasp9 isoform (Figure 5D). Interestingly, F52L and F59L cells from the SW620 line, both untreated cells and the cells treated with TRAIL, showed increased expression of pro-caspase 9, cCasp7, and pro-caspase 6 (*p* < 0.05 according to the Mann–Whitney U test; Figure 5D,F,H, respectively). Moreover, TRAIL increased cPARP in *FUT8*-knockdown clones (Figure 5N; *p* < 0.05 according to the Mann–Whitney U test).

In conclusion, the most conspicuous difference between SW480 and SW620 cells in response to 100 ng/mL TRAIL was the potentiation of the intrinsic mitochondrial pathway—mediated by caspase 9—in SW480 FUT8-knockdown cells, unlike what occurred on their SW620 counterparts.

### 2.5. ERK1/2 Phosphorylation in the FuT-8-Attenuated SW480 Cells Is Independent of TRAIL Exposure

Knockdown of the *FUT8* gene in F52L and F59L clones of the SW480 line increased the expression of phosphorylated ERK1/2 kinase (pERK) in the presence or absence of TRAIL (*p* < 0.05 according to the Mann–Whitney U test; Figure 5M). The TRAIL-induced increase in pERK1/2 expression was greater in the *FUT8*-knockdown clones SW480 F52L and SW480 F59L than in the SW480 wt line. In sharp contrast, in the cell line SW620 panel, the ERK/pERK ratio remained unchanged after *FUT8* knockdown and/or TRAIL treatment, with the exception of a TRAIL-induced pERK increase in the F52L clone (Figure 5N; *p* < 0.05 according to the Mann–Whitney test U).

### 2.6. Concomitant Exposure to TRAIL and Chemotherapy Drugs Enhanced the Inhibition of Cell Proliferation in the FucT-8-Attenuated Cells

To elucidate the involvement of core fucosylation in drug sensitivity of CRC, acute concentrations of different chemotherapeutic and immunotherapeutic agents from cancer pharmacopeia were administered to SW480 and SW620 cells (wt, NTC and F52L/F59L clones). The doses tested were 0.25 μM 5-fluorouracil (5-FU), 0.005 μM oxaliplatin, 0.05 μM irinotecan, 0.005 μM mitomycin C, 100 μg/mL cetuximab, and 100 μg/mL bevacizumab. In addition, to detect possible synergistic effects between TRAIL and chemotherapy agents, SW480 and SW620 cells were grown in parallel under basal conditions (100 ng/mL TRAIL) or in the presence of 100 ng/mL TRAIL and each of the drugs. After 48 h of incubation, cell survival and the rate of inhibition were calculated (Figure 6).

It is noteworthy that TRAIL inhibited the growth of the attenuated SW480 F52L and F59L clones (*p* < 0.05 according to the Kruskal–Wallis test) in all assays performed (upper images in Figure 6A–F), as initially reported (Figure 1A). However, the presence of 5-FU (Figure 6A), oxaliplatin (Figure 6B), irinotecan (Figure 6C), or mitomycin C (Figure 6D) in the culture medium did not lead to a greater inhibition of cell proliferation at the doses tested in the clones F52L/F59L. Similar results were obtained with cetuximab (Figure 6E) and bevacizumab (Figure 6F). In contrast, co-administration of TRAIL and any of the four chemotherapeutics achieved greater inhibition rates, which, in the case of the SW480 F52L clone, were statistically significant compared to those of the NTC clone (*p* < 0.05 according to the Kruskal–Wallis test; Figure 6A–D). Conversely, neither cetuximab nor bevacizumab combined with TRAIL gave rise to synergy in the inhibition of cell proliferation in clones F52L and F59L of the SW480 line (Figure 6E,F). The presence of TRAIL did not affect apoptosis in SW620 F52L/F59L clones (bottom images in Figure 6A–F), as previously reported (Figure 1B). Similarly, the tested anti-tumour agents did not inhibit growth at the assayed concentrations (Figure 6A–F). However, combined with TRAIL, some of these drugs were able to significantly modulate their cytostatic capacity. Specifically, in the SW620 F52L clone exposed to 100 ng/mL TRAIL and 0.05 μM oxaliplatin (Figure 6B) or 0.05 μM irinotecan (Figure 6C), the inhibition rate was significantly reduced in comparison with NTC (*p* < 0.05 according to the Kruskal–Wallis test). Similarly, cotreatment with 100 ng/mL TRAIL and 100 μg/mL bevacizumab significantly reduced the percentage of inhibition of F52L and F59L knockdown clones (*p* < 0.05 according to the Kruskal–Wallis test; Figure 6F).

## 3. Discussion

TRAIL is a type-II transmembrane protein with a domain of high homology to CD95L (Fas ligand) and TNF [43]. As a cytokine of the TNF superfamily, it triggers apoptosis in a cell-type-dependent manner, as in CRC [44], in which resistance to the proapoptotic effect of TRAIL increases throughout malignant progression [45,46]. In this sense, primary SW480 (Dukes B) CRC cells are sensitive to TRAIL and prone to type-II apoptosis [46], while SW620, their isogenic counterparts derived from a metastatic lymph node from the same patient, are considered type-I cells resistant to apoptosis induced by TRAIL [46,47]. The difference in TRAIL sensitivity between SW480 and SW620 cells cannot be attributed to drug-induced resistance, since donor chemotherapy was started after surgery for recurrent cancer [48].

The results of the present study confirmed the development of the TRAIL-resistant phenotype as CRCs progress from poorly metastatic SW480 cells of the primary tumours to highly metastatic regional SW620 cells (Figure 1). Our investigation aimed to go further, specifically regarding the role that core fucosylation may play in the asymmetric response of the SW480/SW620 tandem to TRAIL. For this purpose, we assumed previously established double evidence: (i) glycosylation blockade is associated with overcoming TRAIL resistance in pooled clones of apoptosis-resistant SW480 cells, and (ii) regulation of DR4 is pivotal in driving their positive response to the cytokine [49]. Other background information, such as the positive correlation of TRAIL sensitivity with *FUT3* and *FUT6* mRNA levels [24,50] and the fact that TRAIL resistance originates from *GMDS* gene mutation, which accelerates tumour growth due to evasion of immune surveillance [25], also prompted us to investigate the TRAIL–glycosylation relationship in CRC. Accordingly, to elucidate the relevance of core fucosylation in the response to TRAIL, we analysed the expression of FucT-8 and DR4, as well as the impact of the cytokine on core fucosylation, in wild-type (wt) SW480 cells and the F52L/F59L clones resulting from knockdown of the *FUT8* gene. A similar screening was deployed on SW620 cells to compare the effects of modulating core fucosylation in a TRAIL-resistant CRC line.

In SW480 (Appendix A) and SW620 lines (Appendix A), *FUT8* gene knockdown efficiently reduced mRNA FucT-8 levels, significantly elevating their DR4 expression (Figure 2 and Figure 3). In SW480 F52L/F59L knockdown clones, upregulation of DR4 coincided with their increased cell death ratio in the presence of TRAIL (Figure 1A), unlike in SW620 cells (Figure 1B). Specifically, TRAIL enhanced mitochondrion-regulated apoptosis in *FUT8*-deficient clones of the SW480 line (Figure 5C), whereas their SW620 counterparts lacked active cleaved Casp9 (Figure 5D). Therefore, FucT-8 depletion might be part of the impaired glycosylation that enhanced the sensitivity of the SW480 line to TRAIL-promoted apoptosis [49]. In line with this scenario, previous findings of our team indicating the overactivation of FucT-8 in pre-neoplastic lesions and early CRC stages [51,52] agree with the current observation that core fucosylation protects SW480 primary tumour cells from apoptotic lysis. Furthermore, we recently reported a higher mesenchymal profile and anchorage-independent proliferative capacity in *FUT8*-knockdown clones of SW480 cells than in their syngeneic SW620 counterparts [41].

However, DR4 upregulation alone does not explain the response of SW480 and SW620 lines to TRAIL. This is because F52L and F59L clones from the SW620 line overexpressed DR4 and did not exhibit increased TRAIL-dependent apoptosis (Figure 1B). Consistent with this assumption, depending on the cancer type, in tumour cells expressing DR4 and DR5, both receptors may contribute differently to the enhancement of TRAIL-induced PARP cleavage, as in GMDS-rescued cells [25]. In our experimental conditions, the presence of TRAIL in the culture medium of clones F52L/F59L of the SW480 line increased FucT-8 and core fucosylation (Figure 4C and Appendix A, respectively), maintaining the degree of DR4 upregulation (Figure 4A). Contrarily, TRAIL exposure did not change the expression of FucT-8 and DR4 in F52L and F59L clones of the TRAIL-unresponsive SW620 line (Figure 4B,D). These facts taken together, we consider that core fucosylation might be required to inhibit the apoptotic signal triggered by TRAIL-DR4 recognition, as with SW480 *FUT8*-knockdown clones, although other TRAIL-gathered signalling pathways should be considered as well. In this sense, we focused on the extracellular signal-regulated kinases (ERK1/2 kinases) of the mitogen-activated protein kinase (MAPK) pathway, which control cell growth, differentiation, and apoptosis-inducing signals [53,54,55]. Previous studies have reported that the activation of ERK1/2 via phosphorylation protects against TRAIL-induced cell death [17,53,54,56], and, therefore, MAPK signalling might be involved in TRAIL-dependent apoptosis of SW480 and SW620 cells. Indeed, we provide evidence that TRAIL leads to increased phosphorylation of ERK1/2 in wt SW480 cells (Figure 5M), as opposed to wt SW620 (Figure 5N). Hence, ERK1/2 phosphorylation may function as a dynamic regulator under conditions where SW480 cells must rapidly modulate the apoptotic response, such as in the presence of initiators (i.e., TRAIL). This pro-survival mechanism would compensate for pro-apoptotic signals, making ERK/MAPK signalling a determinant of cell fate and proliferation in CRC [57,58,59]. Furthermore, this important mechanism may be under the control of core fucosylation, as indicated by the significant increase in ERK1/2 phosphorylation (pERK) in *FUT8*-deficient clones F52L and F529L from SW480 cells cultured with or without TRAIL (Figure 5M) and in SW620 F52L clones grown in the presence of cytokine (Figure 5N). Accordingly, we hypothesise that core fucosylation enhances the sensitivity of early CRC to cell death and that, therefore, FucT-8 inhibition could become a strategy to sensitise tumour cells to TRAIL-induced apoptosis. Nevertheless, mechanisms activating death receptors through ERK1/2 have not yet been fully elucidated. In this regard, it has been proposed that FLIP mediates apoptosis by recruiting Raf-1 to the Fas DISC [60,61]. SW480 cells efficiently process FLIP through the TRAIL–DISC complex [62]. So, further research should focus on the role of this inhibitory protein in deregulating apoptotic signals in CRC.

PGP, also known as multidrug resistance protein 1 (MDR1), is an ATP-dependent transmembrane protein that can pump chemotherapy drugs out of a cell. Interestingly, a strong relationship between PGP overexpression and increased sensitivity to both extrinsic and intrinsic TRAIL-induced apoptosis has been found in MDR-transfected breast cancer cells [63]. Moreover, fucosyltransferases have been shown to mediate multidrug resistance in human hepatocellular carcinoma through MDR-related protein 1 (MRP1), another ATP-dependent exportin [64]. In our study, we detected a reverse evolution of FucT-8 and PGP expression, especially in *FUT8*-knockdown clones derived from the SW480 line (Figure 3A,B), suggesting that FucT-8 hyperactivity in early CRC [20,51,65] may enhance resistance to apoptosis and thus contribute to malignant progression [60]. Although the observed variations in PGP expression were not statistically significant, current interest in overcoming chemoresistance and designing patient-tailored cancer therapies [66,67], together with some encouraging results from TRAIL in clinical trials of CRC [68], prompted us to analyse cell viability in response to a combination of TRAIL and some chemotherapy drugs (Figure 6). The drug doses tested in our experimental conditions did not significantly inhibit the growth of wt SW480 and SW620 cells, just as 5-FU did not show any effect on DR4 expression in different CRC lines [69]. Instead, results indicated that the depletion of core fucosylation sensitised SW480 F52L and F59L clones to the combination of TRAIL and low doses of 5-FU, oxaliplatin, irinotecan, or mitomycin C (Figure 6A–D). In contrast, TRAIL combined with oxaliplatin (Figure 6B), irinotecan (Figure 6C), or bevacizumab (Figure 6F) reinforced the resistance to apoptosis exhibited by metastatic SW620 cells. These findings have implications not only for elucidating the role of glycosylation in CRC drug chemoresistance [70] but also for designing therapy formulations that take advantage of TRAIL supplementation and/or modulation of core fucosylation [71]. In one such therapy, the administration of low-dose irinotecan and TRAIL upregulated DR5 expression in TRAIL-resistant HT-29 CRC cells, with the subsequent activation of caspases [72]. Similarly, synergistic potentiation of low-dose 5-FU and TRAIL in TRAIL-resistant gastric adenocarcinoma cells revealed the mediation of DcR2 and DR5 in the activation of caspase cascades [73]. Nonetheless, additional work is needed to identify the role that cell death receptor signalling plays in CRC chemoresistance.

In conclusion, the present study provides evidence that FucT-8 could mediate the TRAIL sensitivity of CRC cells, as well as their response to chemotherapy. In this regard, the activation and/or hyperexpression of FucT-8 in premalignant lesions and early CRC could promote the resistance of, for example, SW480 cells to TRAIL-induced apoptosis and their escape from the surveillance mechanisms that protect tissue homeostasis. However, as malignancy progresses (e.g., in SW480 cells), tumour cells become resistant to cell death and are no longer affected by FucT-8 inhibition. Consequently, modulation of apoptosis sensitivity during early carcinogenesis might be a viable treatment strategy for CRC that is worthy of investigation to address core fucosylated glycan intermediates involved in apoptosis and drug resistance. In this regard, reduced core fucosylation of TNF receptors may be mechanistically linked to decreased mitochondrial-dependent apoptosis because of the activation of the non-canonical NF-κB pathway, as recently reported in osteosarcoma [74]. Therefore, targeting key α(1,6)fucosylated proteins could strengthen the panoply of treatment alternatives for CRC and other neoplasias characterised by exacerbated core fucosylation [75,76].

## 4. Materials and Methods

### 4.1. Cell Culture Conditions

The syngeneic SW480 and SW620 CRC lines obtained from the American Type Culture Collection (ATCC; Manassas, VA, USA) were donated by the Health Research Institute of Santiago de Compostela (IDIS, A Coruña, Spain). Wild-type cell cultures were maintained in high-glucose Dulbecco’s modified Eagle’s medium (DMEM; Sigma-Aldrich, Saint Louis, MO, USA) supplemented with 10% inactivated foetal bovine serum (FBS, Life Technologies, Grand Island, NY, USA) and 10,000 U/mL penicillin–streptomycin (Life Technologies) in a humidified incubator with 5% CO_2_ in air. The cell culture media for attenuated (F52L and F59L) and non-targeted control (NTC) cell lines were also supplemented with 5 µg/mL puromycin (Sigma-Aldrich). Lyophilised human recombinant TRAIL was purchased from Sigma-Aldrich. Chemotherapy drugs were administered by diluting the stock solution in PBS until the desired concentration was achieved. PBS was employed as blank control, 5-fluorouracil 5 g/100 mL was obtained from Accord (Accord Healthcare Ltd., Harrow, UK), oxaliplatin Kabi 5 mg/mL was obtained from Fresenius Kabi (Bad Homburg, Germany), irinotecan Hospira 20 mg/mL was obtained from Pfizer (Madrid, Spain), mitomycin-C 10 mg was obtained from Inibsa Hospital (Barcelona, Spain), cetuximab Erbitux^®^ 5 mg/mL was obtained from Merck (Serono UK Ltd., Welwyn Garden City, UK), and bevacizumab Avastin 25 mg/mL was obtained from Roche (Applied Science, Indianapolis, IN, USA). All other reagents were of the highest purity commercially available.

### 4.2. shRNA Lentiviral Transfection and Phenotypic Selection of FUT8-Attenuated Cells

The cell model employed in this work was developed previously by our research group [41]. Briefly, 96-well plates containing 1.6 × 10^4^ cells were seeded and incubated overnight at 37 °C in an atmosphere of 5% CO_2_ in a humidified incubator. Next, 110 μL/well of hexadimethrine bromide (Sigma-Aldrich) at a final concentration of 8 μg/mL and 15 μL/well of two lentiviral particles from Sigma-Aldrich containing pLKO.1 plasmids targeting human *FUT8* (MISSION lentiviral Transduction Particles pLKO.1-puro-CMV-TurboGFP, TRCN0000035952, and TRCN00000229959) or a non-targeting control (MISSION^®^ pLKO.1-puro-CMV-TurboGFP, SHC003), also from Sigma-Aldrich, were added. The cells were incubated overnight to allow transfection. Next, the culture medium was replaced with fresh DMEM containing 5 μg/mL of puromycin (Sigma-Aldrich). This medium was replaced every 72 h until resistant clones grew up. After successful lentiviral transfection, phenotypic selection of *FUT8*-knockdown clones was achieved via prolonged exposure to *Lens culinaris* agglutinin lectin (LCA) by supplementing complete medium with 500 μg/mL from Vector Laboratories (Peterborough, UK). Clones were forced to remain in this LCA-containing medium for 7 days, and after this passage, cells were seeded in complete medium without LCA.

### 4.3. Cell Proliferation Assay

Experiments of cell proliferation were carried out in triplicate using Alamar Blue^®^ (AB, ThermoFisher, Waltham, MA, USA) as the developing reagent, following a previously published protocol [77]. Initially, 5 × 10^3^ cells were seeded on a 96-well plate (100 µL/well) in standard culture conditions (5% CO_2_, 37 °C). Next, the cells were treated with 50, 100, or 200 ng/mL of human recombinant TRAIL (Biovision, San Francisco, CA, USA) or only the vehicle phosphate-buffered saline (PBS, Sigma-Aldrich) for 24 h. After 24 h of proliferation, 10 µL of AB was directly added into the culture media at a final concentration of 10%, and the plate was returned to the incubator for 3 h. To detect surviving cells, fluorescence was analysed in a FLUOstar Optima fluorimeter from BMG Labtech (Ortenberg, Germany) at wavelengths of 544 nm for excitation and 590 nm for emission. As a blank control, AB was added to the medium without cells (*n* = 3).

### 4.4. Cell Immunofluorescence Assay

Cells were seeded in triplicate at a density of 1 × 10^4^ cells/mL on glass coverslips, placed in a 24-well cell-culture plate, and grown for 24 h in the conditions previously described. After the cells were attached to the glass, they were washed three times with PBS at 500 µL/well and fixed adding 100 μL ice-cold ethanol for 10 min. Permeabilisation required keeping 200 μL of 0.1% Triton X-100 in PBS for 10 min and subsequent blockade of unspecific binding by incubating it with 200 μL of 1.0% bovine serum albumin (BSA) in PBS for 30 min. Next, the coverslips were incubated for 2 h at room temperature with either 100 µL of anti-FucT-8 antibody (1/15 dilution, 66118-1-Ig) from Proteintech (Chicago, IL, USA) or 100 µL of anti-PGP antibody (1/25 dilution, ab3366) from Abcam (Cambridge, UK) in PBS with 1.0% BSA. Secondary incubation consisted of 100 µL of the green Goat Anti-Mouse IgG-Alexa Fluor^®^ 488 conjugate (dilution 1/100, ab150113) from Abcam diluted in PBS with 1.0% BSA at room temperature for 1 h. Nuclei were stained with 1 μg/mL DAPI (4′,6-diamidino-2-phenylindole) following the manufacturer’s recommendations (Thermo Fisher Scientific). Finally, glass coverslips were rinsed three times in 500 µL of PBS and mounted with Vectachield anti-fade mounting medium from Vector Laboratories. Fluorescence images were taken using an Olympus BX51 binocular microscope equipped with an Olympus DP71 digital camera from Olympus Life Science (Shinjuku, Tokyo, Japan) and analysed with Fiji software (version 2.9.0).

### 4.5. RNA Extraction and Real-Time Quantitative PCR (RT-qPCR)

Total mRNA from the cell cultures was extracted using the High Pure RNA Isolation Kit of Roche, and cDNA synthesis was performed with the MuLV Reverse Transcriptase kit from Applied Biosystems (Foster City, CA, USA), in both cases according to the manufacturer’s indications. RT-qPCR was carried out by means of TaqMan assays in an ABI 7500 Real-Time PCR System. The glyceraldehyde-3-phosphate dehydrogenase (*GAPDH*) gene was used as an internal normalisation control. The primers employed are shown in Appendix A. The results were represented as the fold change in gene expression relative to *GAPDH* gene expression (2^−ΔΔCt^).

### 4.6. Cell Protein Extraction

Cell lysates were obtained via the direct addition of homemade radioimmunoprecipitation assay (RIPA) buffer (0.1% SDS, 150 mM NaCl, 50 mM Tris-HCl, pH 8.5, 0.5% sodium deoxycholate, 1% Nonidet P-40, 2 mM Na_3_VO_4_, 4 mM NaF), supplemented immediately before use with a protease inhibitor cocktail tablet from Roche Life Sciences (Penzberg, Bayern, Germany). Cell lysis was manually aided using a scrapper, and the suspension was kept for 30 min in ice, with soft vortexing at intervals. Next, the suspension was subjected to centrifugation at 10,000× *g* (4 °C) for 10 min, the cell debris and non-solubilised material were removed, and the extract was stored at −20 °C. The protein content was assayed using the BCA method (Sigma-Aldrich) using BSA (Sigma-Aldrich) as the reference.

### 4.7. SDS-PAGE

In all, 10 µg of protein per sample was heated with 4× Laemmli loading buffer for 5 min at 100 °C. SDS-PAGE with 10% polyacrylamide was used in all experiments, and the voltage was settled at 200 V for 50 min or until the blue front eluted. Protein bands were well resolved in the 260–20 kDa range using the BlueStar PLUS Prestained Protein Markers from NIPPON Genetics Europe GmbH (Düren, Nordrhein-Westfalen, Germany). Heavier proteins required 12% polyacrylamide gel instead of 10%. Coomassie brilliant blue (0.1% Coomassie blue R250, 5% glacial acetic acid, 30% methanol) was used for protein staining.

### 4.8. Immunoblot

The cell lysates subjected to SDS-PAGE were subsequently analysed using immunoblotting. The primary antibodies used were anti-Caspase3 (#9665), anti-Caspase 6 (#9762), anti-Caspase 7 (#12827), anti-Caspase 8 (#9746), anti-Caspase 9 (#9508), and anti-PARP (#9542) from the Procaspase Antibody Sampler Kit of Cell Signaling Technology (Danvers, MA, USA). The Anti-pERK1/2 (#4094) and anti-ERK1/2 (#4695) were also from Cell Signaling Technology. The Anti-FucT-8 (66118-1-Ig) was from Proteintech. As secondary antibodies, we used horseradish-peroxidase-labelled anti-mouse-IgG (#7076) or anti-rabbit-IgG (#7074) from Cell Signaling Technology. An equal loading was verified using the anti-β-actin antibody (sc-47778) from Santa Cruz Biotechnology (Dallas, TX, USA). Signals were detected using the enhanced chemiluminescence reaction kit of Clarity Western ECL from Bio-Rad (Hercules, CA, USA).

### 4.9. Lectin Blot

The protein-transferred membrane was blocked with 3% BSA for 1 h at room temperature. Next, the membrane was incubated with biotinylated LCA to specifically detect core fucosylated oligosaccharides. After washing, the membranes were incubated for 1 h with horseradish-peroxidase-conjugated avidin using the EliteVECTASTAIN ABC kit (Vector Laboratories). Enhanced chemiluminescence (ECL) development was performed using the Clarity Western ECL kit (Bio-Rad) according to the manufacturer’s protocol. Chemiluminescence intensity was quantified by means of Image Lab v4.1 software. Plots were created in Microsoft Excel 2013, and IBM SPSS Statistics v26 was used for statistical analyses.

### 4.10. Statistical Analyses

Microsoft Excel 2013 and IBM SPSS Statistics v26 were used for graphical and statistical analyses. The non-parametric Mann–Whitney U test was used to determine the differences between two conditions. For comparing more than two different conditions, as in chemotherapeutic assays, the non-parametric Kruskal–Wallis test was indicated. One-way ANOVA and Fisher’s test were used to analyse the mRNA expression levels calculated with the 2^−ΔΔCt^ fold-change method. Statistical significance was set at *p* ≤ 0.05.

## 5. Novelty and Impact

The study provides evidence of how core fucosylation modulates TRAIL-induced apoptosis in colorectal cancer (CRC). Specifically, in early CRC (i.e., primary tumour SW480 cells), *FUT8* knockdown increases apoptosis, while metastatic CRC (i.e., SW620 cells) becomes more refractory to effects on core fucosylation and TRAIL-mediated apoptosis. Furthermore, core fucosylation depletion sensitises CRC to the combination of TRAIL and chemo/immunotherapy. Therefore, the FucT-8 enzyme is a plausible target for increasing apoptosis and drug response in early CRC.

## Figures and Tables

**Figure 1 ijms-24-11879-f001:**
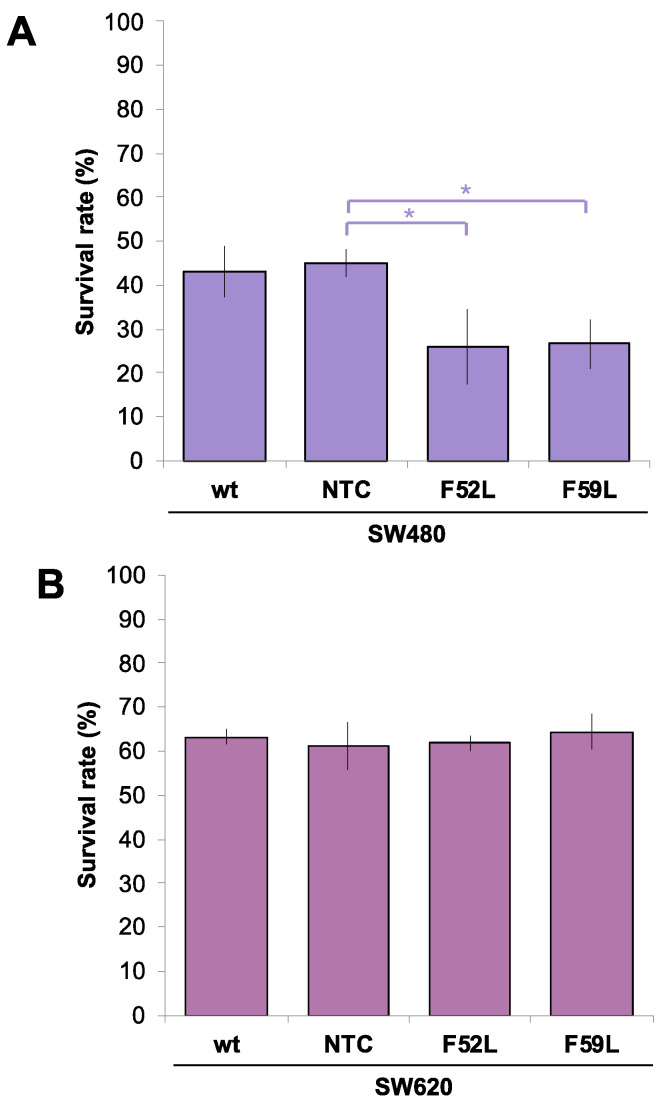
Cell survival rate (%) in the (**A**) SW480 model and (**B**) SW620 model after induction of apoptosis via exposure to 100 ng/mL TRAIL for 24 h. Surviving cells were detected using a fluorimetric assay with Alamar Blue^®^ as the developing reagent. The survival rates (%) were calculated by comparing the cells that survived in the culture with TRAIL with those that survived in the culture without TRAIL. Measurements were taken in three different experiments, and cells were seeded in triplicate. Results are plotted as the mean ± SD, and the NTC clone is used for statistical calculations as the control cell line. The Mann–Whitney U test results were significant at *p* < 0.05 (*). wt: wild-type state; NTC: non-targeted control; F52L and F59L: *FUT8*-knockdown clones selected with *Lens culinaris* agglutinin (LCA).

**Figure 2 ijms-24-11879-f002:**
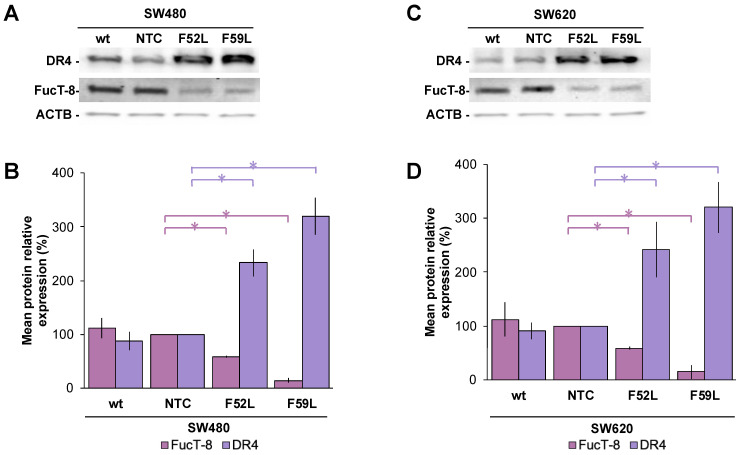
Representative immunoblot of FucT-8 and DR4 protein expression in the (**A**) SW480 model and (**C**) SW620 model. Relative quantitation of FucT-8 (pink bars) and DR4 (violet bars) via chemiluminescence using the β-actin (ACTB) expression as protein loading control in the (**B**) SW480 model and (**D**) SW620 model. Measurements were taken in four different experiments. Results are plotted as the mean ± SD, and the NTC clone is used as the control cell line for statistical calculations. The Mann–Whitney U test results were significant at *p* < 0.05 (*). wt: wild-type cells; NTC: non-targeted control cells; F52L and F59L: *FUT8*-knockdown clones selected with *Lens culinaris* agglutinin (LCA).

**Figure 3 ijms-24-11879-f003:**
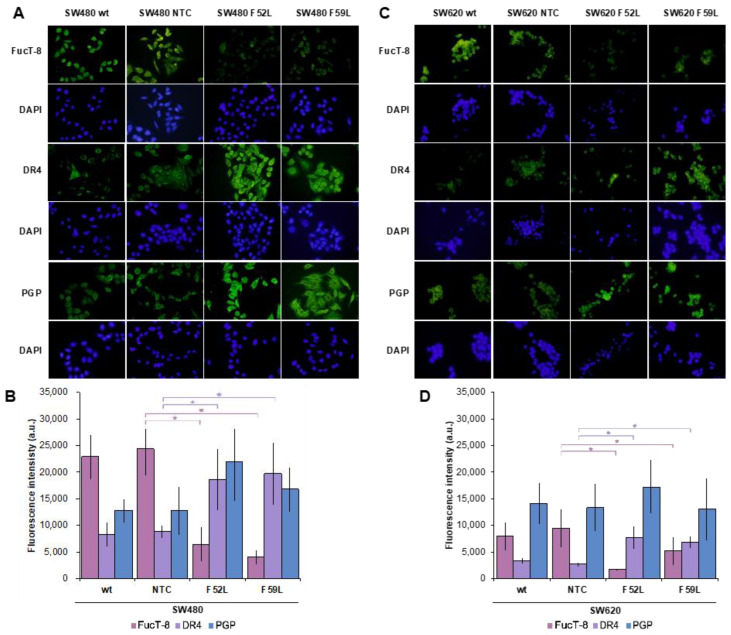
Representative single immunofluorescence of FucT-8, DR4, and PGP protein expression in the (**A**) SW440 model and (**C**) SW620 model. Original magnification ×100 (oil immersion). Quantitation of FucT-8, DR4, and PGP via single immunofluorescence in the (**B**) SW480 model and (**D**) SW620 model using the Corrected Total Cell Fluorescence (CTCF) method. Identical batches of cells were used in three independent experiments. Results are plotted as the mean ± SD, and the NTC clone is used as the control cell line for statistical calculations. The Mann–Whitney U test results were significant at *p* < 0.05 (*). wt: wild-type cells; NTC: non-targeted control cells; F52L and F59L: *FUT8*-knockdown clones selected with *Lens culinaris* agglutinin (LCA).

**Figure 4 ijms-24-11879-f004:**
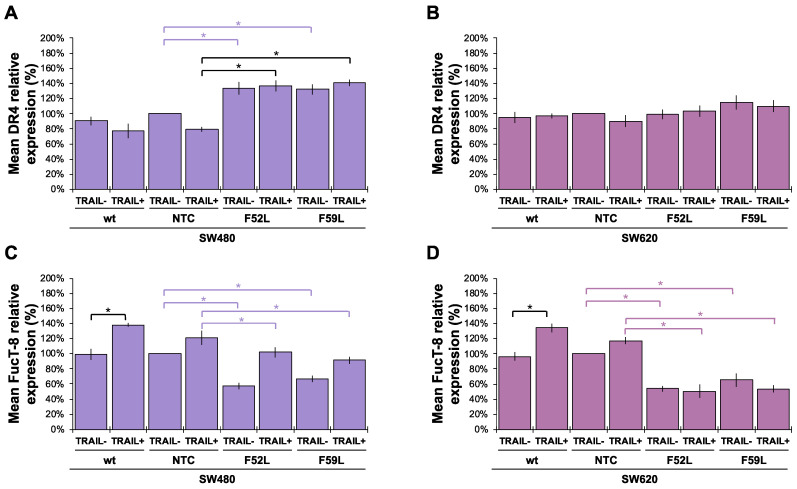
Relative quantitation via immunoblot of DR4 and FucT-8 proteins in the absence (TRAIL-) and presence of 100 ng/mL TRAIL (TRAIL+). DR4 expression in SW480 (**A**) and SW620 (**B**) cells; FucT-8 expression in SW480 (**C**) and SW620 (**D**) cells. Β-actin expression (ACTB) was used as protein loading control and NTC clones as reference. Results are plotted as the mean ± SD. For statistical calculations, NTC clones were used as control. The Mann–Whitney U test results were significant at *p* < 0.05 (*). wt: wild-type cells; NTC: non-targeted control cells; F52L and F59L: *FUT8*-knockdown clones selected with *Lens culinaris* agglutinin (LCA).

**Figure 5 ijms-24-11879-f005:**
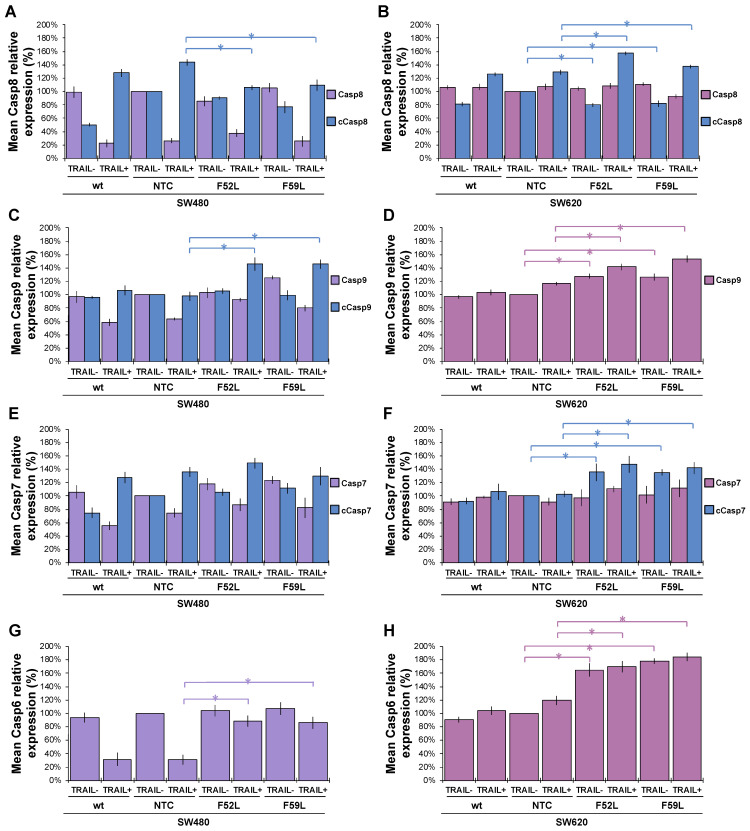
Relative quantitation via immunoblot of apoptosis markers and ERK1/2 MAP kinases in the absence (TRAIL-) and presence of 100 ng/mL TRAIL (TRAIL+). Pro-caspase-8 (Casp8, violet bars) and caspase-8 (cCasp8, blue bars) expression in the SW480 (**A**) and SW620 (**B**) lines; pro-caspase-9 (Casp9, violet bars) and caspase-9 (cCasp9, blue bars) in the SW480 (**C**) and SW620 (**D**) lines; pro-caspase-7 (Casp7, violet bars) and caspase-7 (cCasp7, blue bars) expression in the SW480 (**E**) and SW620 (**F**) lines; pro-caspase-6 (Casp6) expression in the SW480 (**G**) and SW620 (**H**) lines; pro-caspase-3 (Casp3) expression in the SW480 (**I**) and SW620 (**J**) lines; PARP (violet bars) and cleaved PARP (cPARP, blue bars) expression in the SW480 (**K**) and SW620 (**L**) lines; total ERK (violet bars) and phosphorylated ERK1/2 (pERK, blue bars) expression in the SW480 (**M**) and SW620 (**N**) lines. Results are plotted as the mean ± SD. β-actin expression (ACTB) was used as the protein loading control. For statistical calculations, the NTC clone was used as the reference. The Mann–Whitney U test results were significant at *p* < 0.05 (*). wt: wild-type cells; NTC: non-targeted control cells; F52L and F59L: *FUT8*-knockdown clones selected with *Lens culinaris* agglutinin (LCA).

**Figure 6 ijms-24-11879-f006:**
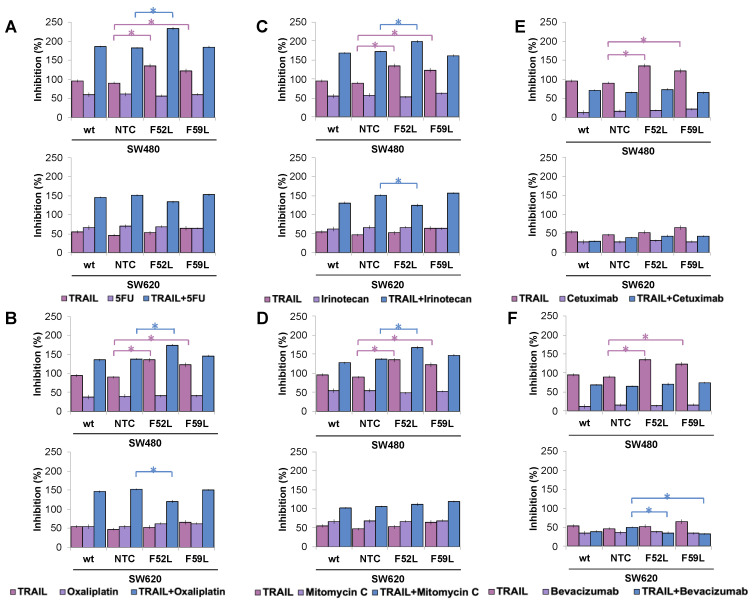
Representative inhibition of cell growth induced by 100 ng/mL TRAIL for 48 h (pink bars), chemotherapy drugs (violet bars), and cotreatment with 100 ng/mL TRAIL and the corresponding chemotherapy agent (blue bars) in the SW480 and SW620 lines. (**A**) Amounts of 0.25 μM 5-fluorouracil (5-FU); (**B**) 0.005 μM oxaliplatin; (**C**) 0.05 μM irinotecan; (**D**) 0.005 μM mitomycin C; (**E**) 100 μg/mL cetuximab; and (**F**) 100 μg/mL bevacizumab. Results (mean ± SD) are from three independent experiments. NTC clones were used as reference for statistical calculations. Inhibition (%) = [(control growth rate − treated sample growth rate)/control growth rate] × 100%. Kruskal–Wallis test was significant at *p* < 0.05 (*). wt: wild-type cells; NTC: non-targeted control cells; F52L and F59L: *FUT8*-knockdown clones selected with *Lens culinaris* agglutinin (LCA).

## Data Availability

The remaining research data are available in the Appendix A.

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
