# Peer review of "Core Fucosylation Mediated by the FucT-8 Enzyme Affects TRAIL-Induced Apoptosis and Sensitivity to Chemotherapy in Human SW480 and SW620 Colorectal Cancer Cells"

_ijms, 2023, doi:10.3390/ijms241511879_

Round 1

Reviewer 1 Report

The study by R.Lopez-Cortes et al. analyzed an important problem in cancer biology, that is, the role of a particular mechanism in the response of tumor cells to death ligands and conventional drugs. The authors compared the two isogenic colon carcinoma cell lines that differ in their metastatic properties. Down-regulation of one single gene responsible for the core fucosylation event differentially affected the survival of each cell line to TRAIL or its combinations with chemotherapeutics. The authors provided a suggestive evidence in support of the involvement of ERKs and caspase 9.

In general, the manuscript is an interesting study performed by the experts in the field. The text is well-written and thoughtful. 

A few questions are supposed to clarify some issues:

1. I am not sure PGP is directly relevant to this story. The authors found no significant differences in this mechanism plus the drugs used for combinations are not transported by PGP. These experiments look more like an unnecessary adjunct although the decision to leave or discard this part is at the discretion of the authors.

2. Figure 6: what is % inhibition on the Y axes? Is it possible to show survival curves (i.e., dose responses) instead of bar diagrams?  

3. Are there data on long-term cell survival? Does FucT-8 confer prolonged survival (14 days or so) or its attenuation gives a temporary/transient effect and can eventually be overcome? 

4. Is there a mechanistic hypothesis why fucosylation is important in early colon cancer whereas at later stages this event becomes dispensable?

5. Are there therapeutic tools to target FucT-8? Is it an established small molecular weight drug target for early colon cancer? 

Generally, the quality of language is good. I suggest shortening some sentences and avoiding unnecessary wording such as 'as per our results'.

The fragment in Introduction (lines 66-81) can be curbed.

Reviewer 2 Report

Please see the attached document for comments.

Minor edits are required.

Round 2

Reviewer 1 Report

The authors have addressed my comments. Now the study can be recommended for publication n IJMS. 

Reviewer 2 Report

Authors have incorporated changes as requested.